# Ergonomic Challenges and Musculoskeletal Pain During Remote Working: A Study of Academic Staff at a Selected University in South Africa During the COVID-19 Pandemic

**DOI:** 10.3390/ijerph22010079

**Published:** 2025-01-09

**Authors:** Nevinia Narainsamy, Francis Fabian Akpa-Inyang, Stanley Chibuzor Onwubu, Nalini Govender, Julian David Pillay

**Affiliations:** 1Department of Chiropractic, Faculty of Health Sciences, Durban University of Technology, Durban 4001, South Africa; narainsamynevinia@gmail.com; 2Faculty of Health Sciences, Durban University of Technology, Durban 4001, South Africa; francisa@dut.ac.za; 3Faculty of Management Sciences, Durban University of Technology, Durban 4001, South Africa; stanleyo@dut.ac.za; 4Department of Basic Medical Sciences, Faculty of Health Sciences, Durban University of Technology, Durban 4001, South Africa; nalinip@dut.ac.za

**Keywords:** musculoskeletal pain (MSP), ergonomics, remote working, COVID-19 pandemic, academic staff

## Abstract

The COVID-19 pandemic led to a rapid shift to remote working, which affected ergonomic conditions and increased the risk of upper body musculoskeletal pain (MSP). This study assessed the prevalence and impact of upper body MSP (affecting the head, neck, shoulders, and back) among academic staff at a University of Technology during the pandemic. Data were collected from 110 participants through an online, descriptive, cross-sectional survey adapted from the Dutch Musculoskeletal Questionnaire, the Standardized Nordic Questionnaire, and the McCaffrey Initial Pain Assessment Tool. The survey examined demographics, ergonomic practices, MSP, and psychological well-being before and during the pandemic. The sample included 59.1% female participants, with most being middle-aged. Persistent MSP was common, with 54.5% reporting neck pain and 59.1% experiencing back pain during the pandemic, alongside a significant decline in wrists in neutral position ergonomics (*p* = 0.012). Psychological well-being also worsened, as participants reported lower levels of cheerfulness, calmness, and energy than before the pandemic. While 81.9% of 90.9% participants with pre-existing MSP continued to experience pain, a small (9%) but notable proportion saw a decline in MSP during this period. These findings highlight a strong relationship between remote working conditions and MSP, with poor ergonomics and psychological distress contributing significantly to persistent pain. The results emphasise the need for institutions to provide ergonomic support, such as appropriate equipment and workstation adjustments, alongside mental health resources to mitigate the long-term impacts of remote working on physical and mental health.

## 1. Introduction

During the COVID-19 pandemic, many organisations implemented remote working as a preventive measure to reduce viral transmission [1,2]. This was a rapid shift, leaving little time to consider ergonomic support for employees. This lack of attention to ergonomics potentially predisposed many workers to musculoskeletal disorders (MSDs) and musculoskeletal pain (MSP), particularly in the head, neck, shoulders, and back regions [3]. MSP is linked to inadequately ergonomically designed equipment, including desks and chairs, commonly used in home offices not designed for extended work hours [4].

An increase in the prevalence of MSDs and MSP was reported during the pandemic, indicative that as individuals transitioned to remote working, experienced heightened discomfort and pain due to suboptimal ergonomic and working environments [4,5]. Moreover, Ekpanyaskul and Padungtod (2021) also reported a significant rise in musculoskeletal complaints, particularly in the neck and back regions, in response to the sudden shift to home offices without ergonomic interventions [6]. These findings corroborate prior data that indicate that poor posture, prolonged sitting, and repetitive movements in inadequately designed workspaces are major contributors to the development of MSP [7,8].

Prior to the pandemic, work-related musculoskeletal disorders (WRMSDs) comprised a significant public health concern in office environments, with long hours of computer use, repetitive tasks, and static postures contributing to the high incidence of MSP [9]. The lockdown restrictions imposed during the pandemic exacerbated this issue, as makeshift workstations such as kitchen tables, sofas, and low-quality chairs lacked the necessary ergonomic support to prevent musculoskeletal discomfort [10]. Additionally, workers who did not have proper ergonomic equipment experienced greater fatigue, discomfort, and pain, which ultimately led to decreased work productivity [11].

The negative impacts of poor ergonomics during remote working conditions are not limited to physical discomfort but also extend to psychological and emotional stress. Recent studies indicate that remote workers, particularly academic staff, faced increased levels of stress and anxiety during the pandemic, exacerbating their physical discomfort [12,13]. The interplay between physical and mental health challenges suggests that remote work during the pandemic contributed to a complex and multifaceted public health issue that demands urgent attention.

MSDs and MSP are associated with repetitive and awkward movements, poor posture, and long hours spent in front of a computer screen without adequate breaks [14]. According to the International Ergonomics Association, poor ergonomic practices can lead to fatigue, pain, and the eventual development of chronic musculoskeletal conditions [15]. These physical issues are compounded when employees lack access to ergonomic chairs, adjustable desks, and proper workstation setups. This consequently affected academics who were required to rapidly shift to online teaching, subsequently amplifying the risk of developing upper body MSP [14].

A study conducted by Alon et al. (2020) echoed these concerns, reporting that nearly 70% of participants who transitioned to remote work experienced new or worsened musculoskeletal symptoms, with the majority reporting pain in their necks, shoulders, and lower backs [16]. The study also highlighted that many remote workers had not received any training on proper ergonomic practices before the pandemic, further increasing their vulnerability to MSP [16]. Bouziri et al. (2020) emphasised the importance of workplace ergonomic interventions, recommending that organisations provide remote workers with the necessary equipment and training to mitigate the risks of MSP [17].

The costs incurred as a consequence of health issues resulting from musculoskeletal disorders (MSDs) due to non-ergonomic working postures can be substantial. MSDs are a major contributor to the global burden of disease, causing significant health loss and disability [18,19]. Studies have shown that work-related musculoskeletal disorders (WRMSDs) can have major financial implications for employees, employers, and industry [20]. The medical costs of MSDs, including healthcare expenditures and lost productivity, have been estimated to account for about 1% to 2.5% of a country’s gross domestic product [21].

However, some studies have shown a decline in musculoskeletal pain (MSP) during remote working [22]. For instance, a recent study by Lin and colleagues [22] found that employees working from home experienced reduced musculoskeletal symptoms compared to their pre-pandemic office-based work. The authors attributed this decline to the increased flexibility and control over their work environment that remote work provided, allowing employees to take more frequent breaks and adjust their posture and workstation setup as needed [22].

Similarly, other research has suggested that the transition to remote work during the COVID-19 pandemic may have had a positive impact on employees’ physical well-being. A study reported that participants with a greater decrease in workplace comfort and ergonomics score during the lockdown had less musculoskeletal pain with an onset during this period [23]. This indicates that the more ergonomic home workstations, compared to the office environment, may have contributed to the reduction in MSP for some remote workers [23].

Academic staff, a population particularly impacted by the pandemic, faced unique challenges related to the sudden shift to remote teaching and administrative duties. According to Bane, Aurangabadkar, and Karajgi (2021), the increased computer usage, combined with awkward sitting postures and limited access to ergonomic equipment, significantly contributed to the high prevalence of MSP among academics [24]. In a similar study, Wang and Zhao (2020) found that academic staff who lacked proper workstation setups were more likely to report severe MSP and a decline in work productivity, underscoring the urgent need for ergonomic interventions in remote working environments [25].

This study aims to determine the prevalence and effects of upper body musculoskeletal pain (MSP) involving the head, neck, shoulders, and back among academic staff at a local University of Technology during the COVID-19 pandemic. It also seeks to examine the relationship between ergonomic conditions and MSP. The findings from this research are expected to inform the development of guidelines and recommendations for preventing MSP in remote working environments and provide valuable insights into the ergonomic considerations needed for future work-from-home scenarios. The study site was conveniently and purposively selected because all the researchers are academic staff members at the institution, and there is limited research on this topic within South Africa.

## 2. Methodology

### 2.1. Ethics Approval and Study Design and Setting

This study was approved by the Institutional Research and Ethics Committee (IREC 136/20). This was a descriptive, cross-sectional study design, utilising an electronic survey questionnaire that was distributed to gather data from academics at a selected University of Technology in South Africa. The questionnaire included sections on basic demographic information and specific questions related to the onset of musculoskeletal pain, as well as the postural and ergonomic effects of remote working during the COVID-19 pandemic.

### 2.2. Study Population

This study was conducted electronically via the university’s internal emailing platform. The population consisted of full-time, non-contract academic staff members from seven (7) university campuses. Participants were recruited based on specific inclusion criteria, i.e., academics who were actively engaged in remote working during the nationwide lockdown imposed in response to the COVID-19 pandemic, as well as academics who continued working remotely after the lockdown. Part-time, temporary, or ad hoc employees, as well as employees not involved in the teaching and learning of undergraduate or postgraduate programs, were excluded.

### 2.3. Sample Size and Participant Recruitment

Approximately 700 academic staff members employed at the university were targeted. Based on a 10% response rate typical for online questionnaire studies, a minimum sample size of 105 participants (n = 105) was determined to be sufficient, in consultation with a statistician.

Participant recruitment was initiated through an advertisement posted on the university’s advertising/communication platform, inviting interested individuals to participate. Prospective participants were asked to contact the researcher via email, after which they were provided with an electronic link containing the letter of information, informed consent form, and questionnaire for completion.

The recruitment and data collection for this cross-sectional study took place between May and November 2021. This timing was just post the COVID-19 peak in South Africa, which occurred between March and December 2020 [26]. The COVID-19 peak in South Africa was during this period, so the timing for the recruitment and data collection was adequate as the experience of the lockdown was still fresh in the minds of the participants [26]. This timing was necessary for the reliability of the data and to factor in any potential recall bias [27].

### 2.4. Data Collection

The data collection tool utilised was a descriptive, cross-sectional survey that was piloted to assess the face validity of the instrument. The tool was created by adapting and modifying three existing online questionnaires: the Dutch Musculoskeletal Questionnaire [28], the Standardized Nordic Questionnaire (SNQ) for musculoskeletal symptoms [29], and the McCaffrey Initial Pain Assessment Tool [30]. Since all questionnaires were publicly available online, no formal permission was required for their use.

The questionnaire underwent review by an expert focus group towards ensuring an organised discussion and interrogation of the questionnaire to enhance the internal validity of the questionnaire. The focus group consisted of the student researcher, two research supervisors, a chiropractic staff member, a chiropractor with relevant research experience, and two participants meeting this study’s inclusion criteria. Subsequent changes were implemented to the questionnaire, resulting in a modified version to be piloted. A pilot study was thereafter conducted on three participants that met this study’s inclusion criteria. These participants were excluded from the possibility of participating in the final study. All comments arising from the pilot study were appropriately reviewed and where necessary used to amend the original questionnaire.

Following receipt of informed consent, the final questionnaire was electronically distributed. The questionnaire comprised seven sections: Section A focused on demographic information; Section B gathered data on the participants’ history of stress; Section C explored medical history; Section D addressed working history prior to the COVID-19 pandemic; Section E examined musculoskeletal pain (MSP) and ergonomics before the pandemic; Section F assessed the work environment during the pandemic; and Section G investigated MSP during the COVID-19 pandemic. Additional recruitment efforts included providing an electronic link that contained the letter of information, the informed consent form, and the questionnaire itself.

Data were collected through online questionnaires administered via the Question Pro platform. The Anti-Ballot Box Stuffing (ABBS) feature on Question Pro was employed to prevent multiple completions of the survey by the same participant, thereby ensuring the integrity of the data. The questionnaires clearly stated the voluntary nature of participation and the intended use of the collected data on the first page, allowing potential participants to make informed decisions about their involvement. Once informed consent was obtained, participants were granted access to complete the questionnaire.

### 2.5. Data Analysis

The statistical analyses were conducted using version 29 of the Statistical Package for the Social Sciences (SPSS, IBM: Armonk, NY, USA). Descriptive statistics was used, where both categorical and continuous data were expressed as frequencies, arithmetic means, and standard deviations (mean ± SD). To evaluate changes in ergonomic conditions and musculoskeletal pain both before and during the COVID-19 pandemic, Wilcoxon signed-rank tests and Fisher’s exact tests were employed. These non-parametric statistical methods were chosen to appropriately handle the nature of the data and to assess differences in paired observations. In addition, the contingency coefficient was selected as the corrected analysis in the statistical measure of the relationship between musculoskeletal pain experienced before and during COVID-19 [31].

Each ergonomic aspect was evaluated based on the number of participants who experienced negative ranks (a decrease in proper ergonomic practice), positive ranks (an improvement in ergonomic practice), and ties (no change). The test statistics, including the Z-value and *p*-value, are used to determine whether the observed changes were statistically significant.

A significance level of *p* ≤ 0.05 was considered statistically significant.

## 3. Results

### 3.1. Demographic Profile

The details of the demographic data and the psychological conditions of the participants in this study are shown in Table 1. Most of the study participants were female (59.1%) and middle-aged, which is the age group 45–54 (30.9%), with an average height of 1.66 m and average weight of 75.33 kg. About half of the participants experienced stress or anxiety before COVID-19, and around one-fifth experienced depression.

### 3.2. Psychosocial Characteristics

Figure 1 presents a comparison of emotional and psychological feelings reported by study participants before and during COVID-19. The measured feelings include being cheerful and in good spirits (FCGS), calm and relaxed (FCR), active and vigorous (FAV), waking up feeling refreshed and rested (WFRR), and having a daily life filled with things that interest them (DLFTTIM). The data suggest that participants generally experienced a decline in positive emotional and psychological states during the COVID-19 pandemic compared to before the pandemic. The notable reduction observed in feelings such as cheerfulness, calmness, activeness, and interest in daily life highlights the pandemic’s significant impact on mental well-being.

### 3.3. Ergonomic Conditions

The frequency of ergonomic practices both pre-COVID-19 and during COVID-19 are shown in Table 2 and Table 3. The prevalence rates for various ergonomic practices, viz., keeping thighs parallel to the floor, supporting feet on the floor or footrest, supporting the back with a backrest, keeping forearms parallel to the floor, and maintaining wrists in a neutral position, did not significantly change from the pre-COVID-19 period to the during-COVID-19 period. The data are suggestive that the participant’s ergonomic practices remained relatively stable despite the changes in work environment and conditions.

Analysis of ergonomic practices before and during COVID-19 showed no statistically significant changes across most postural habits among participants (Table 3). For instance, in maintaining thighs parallel to the floor, 11 participants reported a reduction in maintaining their thighs parallel to the floor, 5 showed increase, and 94 showed no change before and during COVID-19 in maintaining thighs parallel to the floor, resulting in no statistically significant difference (Z = −1.500, *p* = 0.134). Similarly, for feet supported on the floor or footrest, an equal number of participants (11 each) experienced both declines and improvements, while 88 showed no change (Z = 0.000, *p* = 1.000). The analysis for back support by a backrest also revealed no significant change, with 11 participants reporting a decrease, 5 showing improvement, and 94 indicating no change (Z = −1.500, *p* = 0.134). In the case of forearms parallel to the floor, 7 participants showed a decline, 14 showed improvement, and 89 experienced no change, with no significant difference observed (Z = −1.528, *p* = 0.127). However, wrist positioning did show a significant decline, with 15 participants reporting a decrease in neutral wrist positioning, 4 showing improvement, and 91 reporting no change (Z = −2.524, *p* = 0.012).

For other postures, such as keeping shoulders relaxed and not elevated, 11 participants reported a decline, 10 reported improvement, and 89 showed no change, with no significant shift observed (Z = −0.218, *p* = 0.827). Neck positioning in a neutral position showed a trend toward improvement, with 5 participants reporting a decline, 13 showing improvement, and 92 reporting no change, though the result was not statistically significant (Z = −1.886, *p* = 0.059). The tendency to flex the neck while holding the phone showed 12 participants experiencing an increase in flexion, 6 showing improvement, and 92 reporting no change, yielding no significant difference (Z = −1.414, *p* = 0.157). Similarly, head rotation results showed 11 participants reporting an increase, 6 showing improvement, and 93 reporting no change (Z = −1.213, *p* = 0.225). Finally, trunk rotation also showed no significant change, with 3 participants experiencing an increase, 4 showing improvement, and 103 showing no change (Z = −0.378, *p* = 0.705). These findings suggest relative stability in ergonomic practices with the exception of a decline in maintaining wrists in a neutral position during the pandemic.

### 3.4. Musculoskeletal Conditions

The association between musculoskeletal pain before and during the COVID-19 pandemic among a sample of 110 participants is given by the contingency coefficient (Table 4). A statistically significant association was noted between musculoskeletal pain experienced before COVID-19 and the persistence of that pain during the pandemic (*p* = 0.017). The majority (90.9%) of participants who had musculoskeletal pain before COVID-19 continued to experience it during the pandemic (81.9%), while a small proportion of participants experienced decline in pain during COVID-19. The weak association suggested by the contingency coefficient (0.006) might indicate that other factors, possibly related to changes in work environment or lifestyle during the pandemic, could have contributed to the persistence or development of musculoskeletal pain.

The prevalence of different types of musculoskeletal pain (MSPP) reported by participants before and during the COVID-19 pandemic is given in Figure 2. The data indicate a general decrease in the prevalence of musculoskeletal pain (headache (49% versus 26.4%), neck pain (68% versus 54.5%), shoulder pain (57% versus 35.5%), and back pain 72% versus 59.1%)) during the COVID-19 pandemic compared to before. The most significant reduction was observed in headache and shoulder pain, while neck pain and back pain also showed notable decreases, although to a lesser extent.

## 4. Discussion

Our findings provide details on the demographic profile, ergonomic conditions, psychological health, and prevalence of musculoskeletal pain (MSP) amongst a selected UoT population of academics working remotely during the COVID-19 pandemic. The majority of participants were female (59.1%) and middle-aged (30.9%). Approximately half reported experiencing stress or anxiety prior to the pandemic, while 21.8% reported experiencing depression. The findings that psychological distress contributes to the onset and persistence of musculoskeletal pain (MSP) are consistent with existing research highlighting the interconnected nature of mental and physical health [2,8]. Duan and Zhu (2020) suggested that stress and anxiety can exacerbate MSP by increasing muscle tension, reducing pain tolerance, and impacting posture, all of which contribute to the chronicity of pain [2]. The pandemic, with its associated isolation, uncertainty, and increased workloads, has likely amplified these effects, especially for remote workers without adequate ergonomic support. This connection underscores the need for interventions that address both the physical and psychological dimensions of MSP, as psychological well-being appears critical to managing and potentially alleviating chronic pain symptoms.

The psychological impact of the pandemic was profound, with participants reporting a noticeable decline in positive emotional states such as feeling cheerful and in good spirits (56.4% before COVID-19 and 23.6% during COVID-19), calm and relaxed (44.5% before COVID-19 and 21.8% during COVID-19), active and vigorous (48.2% before COVID-19 and 20% during COVID-19), and interested in daily life (53.6% before COVID-19 and 23.6% during COVID-19). This decline aligns with research by Montemurro (2020), who found that the pandemic’s uncertainty, isolation, and disruption to normal life led to widespread psychological distress, including anxiety and depression [12]. The WHO further pointed out in their scientific brief that the COVID-19 pandemic has led to a 27.6% increase (25.1 to 30.3) in cases of major depressive disorder (MDD) and a 25.6% increase (23.2 to 28.0) in cases of anxiety disorders (ADs) worldwide in 2020 [32]. These emotional changes are important because they are strongly associated with physical health outcomes. Psychosocial factors, including stress and depression, have been shown to exacerbate MSP by increasing muscle tension, reducing pain tolerance, and promoting poor posture, all of which contribute to the chronicity of pain [10]. The bidirectional relationship between mental and physical health, particularly during times of crisis such as the COVID-19 pandemic, highlights the need for interventions that address both psychological and physical well-being in remote workers.

In terms of ergonomic conditions, the data suggest that the participants’ ergonomic practices remained relatively stable before and during the COVID-19 pandemic. Key ergonomic behaviours, such as keeping thighs parallel to the floor (78% pre-COVID-19 and 72% during COVID-19), supporting feet on the floor or a footrest (69% pre-COVID-19 and 69% during COVID-19), and maintaining forearms parallel to the floor (44% pre-COVID-19 and 51% during COVID-19), did not show significant changes. This is consistent with the result from Filho and Lucca [33]. This stability in ergonomic practices may be attributed to the nature of the participants’ work or the lack of support from their employers [34,35]. Previous studies have found that companies did not invest in improving the ergonomic conditions of their employees during the pandemic even as the transition to remote work became more prevalent [34,35].

A significant finding we observed is the decline in the maintenance of neutral wrist positions during the pandemic (*p* = 0.012). The deterioration observed in wrist ergonomics is consistent with Bouziri and colleagues, who reported wrist pain as a common issue in remote work settings, particularly for individuals using non-ergonomic keyboards and mice or working from low-quality desks [17]. Improper wrist positioning can lead to conditions such as carpal tunnel syndrome and other repetitive strain injuries, which are exacerbated by prolonged periods of typing without adequate wrist support [9,17]. The observation of significant worsening of wrist ergonomics during the pandemic highlights the need for targeted ergonomic interventions, particularly for individuals working remotely for extended periods.

Interestingly, while some ergonomic practices remained unchanged, other factors, such as neck positioning, showed a trend toward improvement (*p* = 0.059). This trend could be indicative of increased awareness of the importance of posture during remote work, potentially spurred by widespread media coverage and workplace recommendations promoting ergonomic best practices during the pandemic [6]. Despite this awareness, significant improvements in ergonomic practices were not widely observed, suggesting that awareness alone may be insufficient to drive behavioural change without the provision of proper ergonomic tools and training.

Our findings also highlight a significant association between pre-existing MSP and its persistence during the pandemic (*p* = 0.017), which is consistent with previous studies [14,36]. Gomez et al. (2023) found that individuals with pre-existing musculoskeletal conditions were more likely to continue experiencing pain under the suboptimal ergonomic conditions associated with remote work [36]. Moreover, 90.9% of participants who reported MSP before the pandemic continued to experience it during the pandemic, underscoring the chronic nature of MSP. This persistence of pain may be explained by the lack of ergonomic interventions and the continued use of improper workstations during the pandemic, as well as the exacerbating effects of psychological stress, which has been shown to prolong and intensify pain symptoms [14].

While the overall prevalence of MSP remained high, this study did observe a general decrease in certain types of MSP during the pandemic, particularly headaches, neck pain, shoulder pain, and back pain. The most significant reductions were seen in headaches (49% pre-COVID-19 vs. 26.4% during COVID-19) and shoulder pain (57% pre-COVID-19 vs. 35.5% during COVID-19). These findings are somewhat unexpected, given the increased sedentary behaviour and the poor ergonomic conditions reported by many remote workers. However, they may be explained by the reduced commute times and increased control over work schedules that remote working offers. Previous studies, such as those by Wang and Zhao (2020), have suggested that remote workers often take more frequent breaks, have more flexibility in adjusting their postures, and can manage their workloads more autonomously, all of which could contribute to a reduction in certain types of MSP [25]. Additionally, the decrease in headache prevalence may be attributed to the reduced exposure to environmental stressors such as noise and lighting in traditional office settings, as well as decreased stress from commuting [24].

Despite these reductions, the high prevalence of MSP during the pandemic, particularly among those with pre-existing conditions, underscores the need for long-term strategies to address the ergonomic and psychological challenges associated with remote working. The findings of this study suggest that future efforts should prioritise providing remote workers with ergonomic tools and training to enhance their knowledge and improve workstation setups. Additionally, mental health support should be made available to help mitigate the impact of stress on physical well-being [37]. These interventions could help prevent the development of chronic MSP and improve the overall quality of life for remote workers.

This study makes a valuable contribution to the growing body of research on the effects of remote working during the COVID-19 pandemic, particularly by highlighting the significant impact of poor ergonomics and psychological stress on musculoskeletal health. The findings underscore the urgent need for comprehensive ergonomic and mental health interventions to support remote workers, especially as remote work continues to be a prevalent mode of employment post-pandemic, both globally and in South Africa.

This study’s timeliness and relevance are notable, as it was conducted during the height of the pandemic, a time when many employees were forced to adapt to remote working conditions with little preparation [1]. By focusing on upper body musculoskeletal pain (MSP) among academic staff, a group that was significantly affected by the sudden shift to online teaching, this research provides critical insights into the ergonomic and mental health challenges faced by this population. This focus on academic staff is especially important in South Africa, where many universities rapidly transitioned to online learning, creating new stressors and physical demands on educators. A key strength of this study lies in the comprehensive data collection tool used, which was adapted from validated instruments like the Dutch Musculoskeletal Questionnaire and the Standardized Nordic Questionnaire (SNQ). This ensured that detailed, reliable data on ergonomic practices and MSP were gathered, enhancing this study’s validity. Additionally, this study’s holistic approach, examining both ergonomic conditions and psychosocial well-being, allows for a more complete understanding of how the shift to remote work has affected both physical and mental health. This dual focus is crucial for designing interventions that address the full range of challenges faced by remote workers.

The use of an online data collection platform further facilitated accessibility, especially during the pandemic’s lockdown restrictions. The Anti-Ballot Box Stuffing (ABBS) feature of the platform helped maintain data integrity by preventing multiple submissions, while the inclusion of academic staff from various campuses ensured a diverse representation of the population under study. However, a key limitation is the cross-sectional design, which limits its ability to establish causal relationships or assess long-term trends in MSP and ergonomic practices. Given that remote working has continued and evolved even after the initial pandemic lockdowns, a longitudinal approach might have offered more comprehensive insights into how MSP and ergonomic behaviours change over time. The lack of exploration into the availability or effectiveness of ergonomic interventions is another limitation. Understanding whether participants had access to ergonomic equipment or training and how these resources impacted their experiences would have provided valuable context. Further, this study’s focus on academic staff at a University of Technology may limit the applicability of its findings to other professional groups and academic staffs in other institutions. While academics faced unique challenges during the pandemic, remote workers in other fields may have had different experiences and ergonomic issues that were not captured in this study. Future research could expand on these findings by including a broader range of professions to better understand the diverse impacts of remote work across sectors in South Africa and beyond. Further investigation is also needed to clarify the specific relationship between mental health and MSP, exploring how psychological stress may impact musculoskeletal health through physiological mechanisms. In addition, this study was conducted during COVID-19, the data reflect both the pandemic and remote working, and the effects of both were not separable. A future study of remote work settings without the pandemic as the background should be performed to further test the impact of remote working on MSP. Thus, while this study provides important insights into the ergonomic and psychological challenges of remote work among academic staff, its limitations highlight the need for further research to broaden the scope and deepen the understanding of MSP in different remote working contexts.

### Recommendations

Our findings warrant the need for institutions to implement ergonomic interventions to reduce the risk of MSP among remote workers. Employers should provide access to ergonomic equipment such as adjustable chairs, proper desks, and keyboard supports, alongside training on how to optimise home workstations. Regular ergonomic assessments and resources, including video tutorials and online ergonomic consultations, as well as addressing mental health challenges, could help remote workers establish healthier working environments. Institutions should offer mental health support through counselling services, stress management workshops, and virtual social activities that promote connection among staff members, which may help in mitigating the psychological toll of working in isolation, which has been shown to exacerbate physical pain.

Further longitudinal design investigations are warranted in other population types to explore the long-term effects of remote working on MSP as well as provide insights into how ergonomic practices and MSP evolve over time, especially as remote working becomes more entrenched in the workforce. Expanding the research to a wider range of occupations would also help identify different ergonomic needs and challenges across various work sectors.

## 5. Conclusions

Our findings highlight the significant impact of remote working during the COVID-19 pandemic on the upper body musculoskeletal pain (MSP) and ergonomic practices of academics at a selected University of Technology. The findings show that while certain ergonomic practices remained stable, there was a marked decline in key areas such as wrist ergonomics, which contributed to increased discomfort and pain. The persistence of MSP, particularly among individuals with pre-existing conditions, underscores the chronic nature of these issues when proper ergonomic interventions are not in place. Additionally, this study reveals a notable decline in psychological well-being, with participants reporting higher levels of stress and anxiety during the pandemic, further aggravating their physical symptoms. These findings underscore the interconnected nature of physical and mental health in the context of remote work. The insights gained from this research point to a growing need for workplace strategies that prioritise both ergonomic support and mental health. As remote working continues to be a significant part of the modern work environment, it is crucial for institutions and employers to address the long-term implications of improper work setups and the mental strain associated with isolation and increased screen time.

## Figures and Tables

**Figure 1 ijerph-22-00079-f001:**
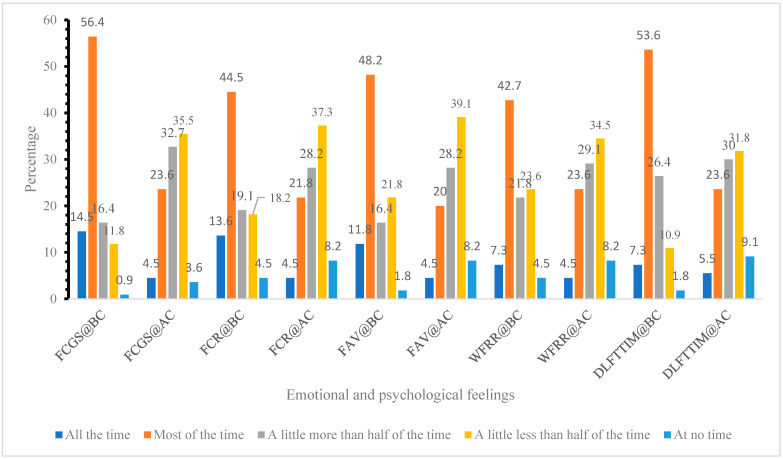
Emotional and psychological feelings express by the study participants before and during COVID-19 (n = 110). (Notes: I have felt cheerful and in good spirits (FCGS); I have felt calm and relaxed (FCR); I have felt active and vigorous (FAV); I woke up feeling refreshed and rested (WFRR); My daily life has been filled with things that interest me (DLFTTIM).) (BC: before COVID-19 onset; AC: after COVID-19 onset.

**Figure 2 ijerph-22-00079-f002:**
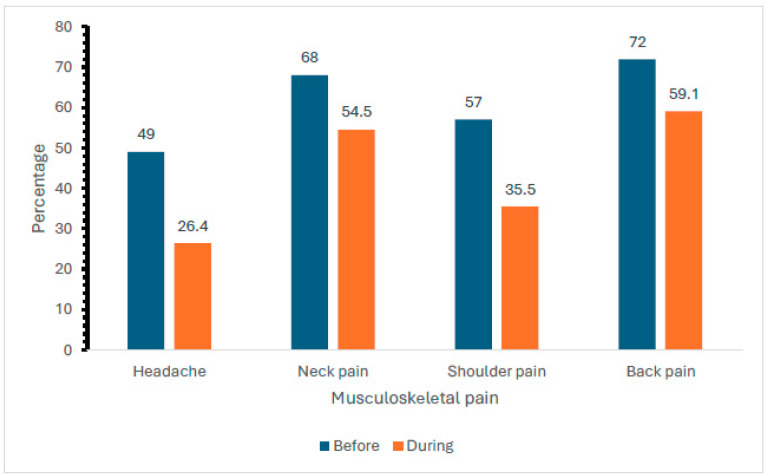
The prevalence of different types of musculoskeletal pain (MSPP) reported by participants before and during the COVID-19 pandemic.

**Table 1 ijerph-22-00079-t001:** The demographic data and the psychological conditions of the study participants before and during COVID-19 (n = 110).

Characteristics	n (%)
**Gender**	
Female	65 (59.1)
Male	44 (40)
Other	1 (0.9)
**Age group**	
25–34	20 (18.2)
35–44	30 (27.3)
45–54	34 (30.9)
55–64	21 (19.1)
>64	5 (4.5)
Height mean ± SD [1.66 ± 0.21]	
Weight mean ± SD [75.33 ± 19.07]	
**Stress/anxiety pre-COVID-19**	
Yes	50 (45.5)
No	60 (54.5)
**Depression pre-COVID-19**	
Yes	24 (21.8)
No	86 (78.2)

**Table 2 ijerph-22-00079-t002:** The frequency of ergonomics health effect occurrence patterns, a prevalence rate with a 95% confidence interval (CI), and trend comparisons pre- and during COVID-19.

Ergonomics (n = 110)	Frequency of the Occurrence Pattern		Overall Prevalence Rate (95% CI)	*p*-Value
	Pre-COVID-19	During COVID-19		
Thighs parallel to the floor	78	72	68.2% (0.904–1.298)	0.469
Feet supported on the floor or footrest	69	69	62.7% (0.816–1.226)	1.000
Back supported by a backrest	45	39	38.2% (0.823–1.617)	0.488
Forearms are parallel to floor	44	51	43.2% (0.636–1.170)	0.414
Wrists in neutral position	64	53	53.2% (0.940–1.551)	0.177
Shoulders relaxed at all times and not elevated	36	35	32.3% (0.701–1.508)	1.000
Neck in neutral (i.e., chin is level)	32	40	32.7% (0.546–1.173)	0.315
Spend a significant amount of time with your neck flexed while holding the phone	29	23	23.7% (0.788–2.054)	0.344
Spend a significant amount of time with your head rotated	42	37	35.9% (0.797–1.618)	0.574
Spend a significant amount of time with your trunk rotated	15	16	14.1% (0.488–1.801)	1.000

Note: “Spend a significant amount of time with your neck flexed while holding the phone” refers to the prolonged posture of bending your neck forward or downward to look at a phone or mobile device. Overall prevalence = total prevalence rate measured for all participants.

**Table 3 ijerph-22-00079-t003:** Results of Wilcoxon signed-rank test comparing ergonomic postures before and during COVID-19.

Ergonomics (n = 110)	Negative Ranks			Positive Ranks			Test Statistics		
	n	Mean Rank	Sum of Ranks	n	Mean Rank	Sum of Rank	Ties	Z	*p*
Thighs parallel to the floor (before–during COVID-19)	11	8.5	93.5	5	8.5	42.5	94	−1.500 ^b^	0.134
Feet supported on the floor or footrest	11	11.5	126.5	11	11.5	126.5	88	0.000 ^b^	1.000
Back supported by a backrest (before–during COVID-19)	11	8.5	93.5	5	8.5	42.5	94	−1.500 ^b^	0.134
Forearms are parallel to floor (before–during COVID-19)	7	11	77	14	11	154	89	−1.528 ^a^	0.127
Wrists in neutral position (before–during COVID-19)	15	10	150	4	10	40	91	−2.524 ^b^	0.012 *
Shoulders relaxed at all times and not elevated (before–during COVID-19)	11	11	121	10	11	110	89	−0.218 ^b^	0.827
Neck in neutral (i.e., chin is level) (before–during COVID-19)	5	9.5	47.5	13	9.5	123.5	92	−1.886 ^a^	0.059
Spend a significant amount of time with your neck flexed while holding the phone (before–during COVID-19)	12	9.5	114	6	9.5	57	92	−1.414 ^b^	0.157
Spend a significant amount of time with your head rotated (before–during COVID-19)	11	9	9.9	6	9	54	93	−1.213 ^b^	0.225
Spend a significant amount of time with your trunk rotated (before–during COVID-19)	3	4	12	4	4	16	103	−0.378 ^a^	0.705

* Indicates statistically significant change significant at *p* ≥ 0.05. ^a^ Based on negative ranks. ^b^ Based on positive ranks.

**Table 4 ijerph-22-00079-t004:** Association between musculoskeletal pain experienced before and during COVID-19.

			Musculoskeletal Pain During COVID-19		Total	Fisher’s Exact Test	Contingency Coefficient
			Yes	No			
Musculoskeletal pain before COVID-19	Yes	Count	85	15	100	0.017	0.006
		% within MSPP	85.0%	15.0%	100.0%		
		% within IDMSP	94.4%	75.0%	90.9%		
	No	Count	5	5	10		
		% within MSPP	50.0%	50.0%	100.0%		
		% within IDMSP	5.6%	25.0%	9.1%		
Total		Count	90	20	110		
		% within MSPP	81.8%	18.2%	100.0%		
		% within IDMSP	100.0%	100.0%	100.0%		

## Data Availability

The data presented in this study are available on request from the corresponding author. The data are not publicly available due to ethical reasons.

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
