# Peer review of "Ergonomic Challenges and Musculoskeletal Pain During Remote Working: A Study of Academic Staff at a Selected University in South Africa During the COVID-19 Pandemic"

_ijerph, 2025, doi:10.3390/ijerph22010079_

Round 1
Reviewer 1 Report
Comments and Suggestions for Authors
In the abstract, specify the participants. It is unclear what 'Strong Link' means in the statement (lines 26-27) .
Methodology:
It is mentioned that 105 participants were sufficient for data collection, but why was the number of participants increased to 110? The pretesting of the questionnaire among three participants was insufficient to finalize its content. It should be clarify what 'All statistical analyses were performed' (Line 167) refers to? The year when data was collected should be specified. It should be mentioned out how the data on MSP, ergonomic factors, and environmental factors were evaluated before and during the pandemic. (lines 153-155)
Results:
Table 1 should contain information about the participants' job designation or position as academics and length of job as academics. It is preferable to provide demographic data and the psychological conditions in separate tables. In Table 2 the average duration of the each working posture should be specified. Provide an operational definition of the term 'significant amount.' Why 'thighs parallel to the floor' is highlighted? The title of the column, 'overall prevalence rate,' is unclear. The table number should be included in the description of findings in lines 220 to 247. Decline, improvement and no change of what are unclear (lines 222-223). Table 4 shows a higher level of pain, while Figure 2 shows a decrease in pain during COVID-19, is confusing. The results section includes comments on the findings, which should be included in the discussion section.
Reviewer 2 Report
Comments and Suggestions for Authors
I enjoyed reading your manuscript. I truly appreciate your hard work in conducting this research. The topic presented is quite interesting and can serve as a basis for improvements in the future. However, several points require further explanation, or the placement of information needs to be adjusted to enhance clarity and make the manuscript more coherent.

Reviewer 3 Report
Comments and Suggestions for Authors
The study examines the relationship between remote work during the COVID-19 pandemic and upper-body musculoskeletal pain (MSP), a topic of practical significance that urgently requires attention. The physical and mental health impacts of remote work warrant in-depth investigation. Data were collected through a descriptive cross-sectional survey using reputable tools such as the Dutch Musculoskeletal Questionnaire and the Standardized Nordic Questionnaire. The choice of these tools enhances the reliability of the data.
Areas for Improvement:
1.Although the sample of 110 participants provides support for the preliminary analysis, it is relatively limited and concentrated. Future research should expand the sample size and include different types of institutions or industries to improve the generalizability and representativeness of the findings.
2.The study mentions demographic and mental health factors but does not sufficiently explore other potential influences. A more detailed multivariate analysis is recommended to control for confounding variables.
3.Further investigation is needed to clarify the specific relationship between mental health and MSP, exploring how psychological stress may impact musculoskeletal health through physiological mechanisms.
Comments on the Quality of English LanguageThe study examines the relationship between remote work during the COVID-19 pandemic and upper-body musculoskeletal pain (MSP), a topic of practical significance that urgently requires attention. The physical and mental health impacts of remote work warrant in-depth investigation. Data were collected through a descriptive cross-sectional survey using reputable tools such as the Dutch Musculoskeletal Questionnaire and the Standardized Nordic Questionnaire. The choice of these tools enhances the reliability of the data.
Areas for Improvement:
1.Although the sample of 110 participants provides support for the preliminary analysis, it is relatively limited and concentrated. Future research should expand the sample size and include different types of institutions or industries to improve the generalizability and representativeness of the findings.
2.The study mentions demographic and mental health factors but does not sufficiently explore other potential influences. A more detailed multivariate analysis is recommended to control for confounding variables.
3.Further investigation is needed to clarify the specific relationship between mental health and MSP, exploring how psychological stress may impact musculoskeletal health through physiological mechanisms.
Reviewer 4 Report
Comments and Suggestions for Authors
Review report for IJERPH -3337829: Ergonomic Challenges and Musculoskeletal Pain during Remote Working: A Study of Academic Staff at a Selected University in South Africa During the COVID-19 Pandemic
While this manuscript reports an interesting and timely study, and the finding is very useful with impact, the authors did not know how to position their findings in the literature, and did not set up a proper framework to effectively disseminate their results. The introduction fails to prepare for the results and therefore it is not possible to have a proper Discussion.
1. Line 35: “coerced” does not seem to be an appropriate word as it is quite subjective. This can easily be rephrased to “… organisations implemented remote working…” without any judgment to the order.
2. Lines 37-40: the use of reference 3 is very weak, as you found out in your study that many workers had existing MSP before the pandemic, and some may experience decreased MSP. Using it will be against your own finding. That is, the logic of this sentence does not hold well.
3. For the second paragraph, you should consider alternative evidence that MSPs did not increase over that period of time. Here is one example: https://doi.org/10.1016/j.ergon.2024.103653. Presenting alternatives in Introduction lays the groundwork to support your own findings.
4. This is the weakest part of this manuscript. All the paragraphs in Introduction are set to ascertain the adverse effects of remote work, mentally or physically. These paragraphs provide strong justification and hypothesis that the study sets to demonstrate those adverse effects, and lead readers to those expectations. The remote work, or work from home has possible benefits such as work-life balance, flexible hours/breaks among others. The authors did not try to provide an alternative possibility for the MSP remaining stable or decline in a remote setting. Therefore, as shown by the results that MSP did not increase, either stayed stable or declined, there is no strong counter arguments in Discussion. Many articulations are guessed, without strong logics, such as point xx below.
5. There is no data administration or collection time (month/year) in 2.3. When did the study take place? It is relevant as this is a cross-sectional study, and part of the survey was to ask the respondents to “recall” before the pandemic (how much time was passed by then?), or even at the peak of the pandemic (when was it locally that the pandemic reached the peak)? Some countries returned to normal sooner than others. Your data collection needs to be placed relative to the events felt by you (or the participants). Provide references of recall bias and reliability and demonstrated that your timing was acceptable.
6. Because the survey was modified from existing scales (2.4), you need to provide the survey as an appendix for evaluation and future reference for comparison or reproduction. Those descriptions on lines 150-157 are useless without the actual questions.
7. Missing reference information about Question Pro platform (line 159). Or that ABBS feature. This is not assessable as it is written.
8. Similar to comment 6 above, without seeing the questions and possible answers, the whole section of 2.5 Data Analysis is meaningless.
9. Line 187: middle aged is not defined. Do not repeat information/numbers available in Tables.
10. The color scheme of Figure 1 does not make sense. If on a scale, the color pattern should follow the frequency scale. It is impossible to comprehend the bars in this figure. If you want to compare a certain measure before and after COVID, they should be presented next to each other, similar to Figure 2.
11. This statement on lines 313-314 “This stability may be attributed to the fact that many remote workers lacked access to proper ergonomic equipment or guidance…” has a huge logical error. An easier explanation is that the nature of their work, or the support from their employers did not exist or is lacking. So it does not matter if they work on site or remotely, their ergonomics conditions are insufficient. This is more logical than your attribution of lacked access during remote settings.
12. Paragraph starting line 340: again I doubt that you can make this conclusion, as 100 of your 110 participants had high MSP before the pandemic, this is not an equal sample problem and can hardly draw any correlation.
13. The most useful information to me is the paragraph starting line 350. However, due to the lack of effort in Introduction, the authors do not quite know how to properly explain this finding. What is this is expected, as explained by reference 19 and many other studies before? The authors should learn how to challenge existing believes that turn out to be false.
14. Line 407: the authors never present clearly what the main differences are with academics compared with other groups being studied before. There is a lack of comparison elsewhere in this paper, so this limitation is self-imposed. If it was properly addressed in Introduction and Discussion, this is not a limitation.
15. A real limitation was that the study was conducted during COVID (while no timing information was ever provided as raised in comment #5), the data may reflect both the pandemic and remote working, or that the effects of both were not separable. A future study of remote work settings without the pandemic as the background should be done.
Due to these critical lacking of proper framework, the recommendations and conclusion are not assessed. The authors should revisit the literature, and present this study and the results as they are, and highlight the important, significant findings that the MSP decreased when people work remotely. The whole paper needs a new framework and organization.
Round 2
Reviewer 1 Report
Comments and Suggestions for Authors
Thank you for responding to my comments. I recommend the manuscript be considered for publication.